# Polyurethane Foam Residue Biodegradation through the *Tenebrio molitor* Digestive Tract: Microbial Communities and Enzymatic Activity

**DOI:** 10.3390/polym15010204

**Published:** 2022-12-31

**Authors:** Jose M. Orts, Juan Parrado, Jose A. Pascual, Angel Orts, Jessica Cuartero, Manuel Tejada, Margarita Ros

**Affiliations:** 1Departament of Biochemistry and Molecular Biology, Facultad de Farmacia, Universidad de Sevilla, C/Prof. García Gonzalez 2, 41012 Sevilla, Spain; 2Department of Soil and Water Conservation and Organic Waste Management, Centro de Edafologia y Biología Aplicada del Segura (CEBAS-CSIC), University Campus of Espinardo, 30100 Murcia, Spain; 3Grupo de Investigacion Edafologia Ambiental, Departamento de Cristalografia, Mineralogia y Quimica Agricola, E.T.S.I.A. Universidad de Sevilla, 41004 Sevilla, Spain

**Keywords:** plastic, mealworms, insect, bacteria, gut microbiome

## Abstract

Polyurethane (PU) is a widely used polymer with a highly complex recycling process due to its chemical structure. Eliminating polyurethane is limited to incineration or accumulation in landfills. Biodegradation by enzymes and microorganisms has been studied for decades as an effective method of biological decomposition. In this study, *Tenebrio molitor* larvae (*T. molitor*) were fed polyurethane foam. They degraded the polymer by 35% in 17 days, resulting in a 14% weight loss in the mealworms. Changes in the *T. molitor* gut bacterial community and diversity were observed, which may be due to the colonization of the species associated with PU degradation. The physical and structural biodegradation of the PU, as achieved by *T. molitor*, was observed and compared to the characteristics of the original PU (PU-virgin) using Fourier Transform InfraRed spectroscopy (FTIR), Thermal Gravimetric Analysis (TGA), and Scanning Electron Microphotography (SEM).

## 1. Introduction

Worldwide plastic polymer production and consumption are increasing every year, reaching around 350–400 million tons in 2019. Its reduction and elimination are a huge challenge [1] since an estimated 2.1 to 3.6 million tons are generated per year in Europe [2]. Polyurethanes (PU) are a type of plastic polymer used as foam in automobile seats, coatings, sealants, the textile industry and other areas [3], with around 4 million tons produced worldwide in 2019 [1]. PU foam is synthesized by the reaction of Diisocyanate (R–N=C=O) and Polyol (R’–OH), producing organic units called urethane [4]. Numerous groups of urethane joined together form a polyurethane molecule [5,6]. Polyurethanes are characterized as resistant to physical and biological degradation due to their chemical composition, which is highly resistant to temperatures and hydrophobicity, resulting in their long lifespans [7].

Recycling polyurethane foam (PU foam) is complicated [8], although mechanical transformation processes, such as crushing and compression molding or pulverizing, can be employed [9]. PU foam waste can also be used as a filler load in lower-value products [10]. However, not enough PU foam is used this way, and the large amount of accumulated PU waste is a big environmental problem. This PU waste is usually incinerated, and the gas emitted in this process contributes to the greenhouse effect. Potentially toxic gases are also emitted from polyurethane combustion, and that not burned accumulates in landfills or aquatic systems [8,11].

New approaches to PU biodegradation using enzymes, e.g., cholesterol esterases, proteases, lipases [12,13], or microorganisms, e.g., *Cladosporium pseudocladosporioides* or *Paracoccus* sp. [7,14] that attack the PU bonds are being investigated, with interesting degradation results. The use of insects to biodegrade plastic is a new natural approach [15,16]. During the last ten years, *Tenebrio molitor* larvae (*T. molitor*; yellow mealworms), commercially used as animal feed and potential alternative protein for human consumption [17,18], have been reported to ingest and degrade different types of plastics, i.e., polyethylene (PE), polystyrene (PS), or the less-studied polyurethane (PU) [15,16,19]. Studies have shown that *T. molitor*’s ability to biodegrade plastics is mainly dependent on its intestinal microorganisms and adaptation to different foods [20,21].

The purpose of this study was to examine: (i) the feasibility of biodegradation, the chemical and physical changes in PU foam using the mealworm larvae of *T. molitor*; and (ii) the gut enzyme activity and microbiota changes associated with feeding polyurethane to *T. molitor*.

## 2. Materials and Methods

### 2.1. Plastics and Mealworms

The polyurethane foam (PU) used in this experiment was obtained from Interplasp S.L. (total Carbon 540.8 ± 15.6 g kg ^−1^; total Nitrogen 43.6 ± 1.09 g kg^−1^; total Phosphorous 0.0037 ± 0.0016 g kg^−1^; total Potassium 0.1 ± 0.02 g kg^−1^). An Inductively Coupled Plasma (ICP) analysis was carried out to check for the presence of heavy metals in the polyurethane composition that could be toxic to *T. molitor* larvae, such as Cd, Hg, or Pb [22].

The mealworm larvae of *Tenebrio molitor* (*T. molitor*) were purchased from Proteinsecta (Albacete, Murcia). Before starting the experiment, the mealworms were fed wheat bran except for the last 24 h, when all food was withdrawn. They were fed PU or wheat bran (bran) for 17 days, with the food incorporated at the beginning of the experiment and not re-established. Three replicate containers with 100 g of randomly selected mealworms were fed 15 g of PU foam, as the PU diet (PU), and three replicate containers of 100 g of randomly selected mealworms were fed 100 g of wheat bran, as the control diet (bran). The containers were incubated in the dark at 27 ± 1 °C and with 80 ± 3% relative humidity. This made it possible to determine the behavior of the microbiota and metabolism of the mealworms when they did not obtain any additional nutrients.

### 2.2. Analysis of PU Foam Biodegradation

#### 2.2.1. Polyurethane Consumption

To evaluate the PU foam biodegradation, the PU and mealworms were sampled at 3, 10, and 17 days after feeding the mealworms with PU and wheat bran. At each sampling, the mealworms, PU, and frass (feces) were separated and stored. The equivalent of 5 g of mealworms was taken out and frozen for further analysis. The PU foam and the mealworms were weighed to calculate the mealworms’ weight loss (%) and PU consumption (%).

#### 2.2.2. Analysis of the PU Foam with Fourier Transform InfraRed Spectroscopy (FTIR)

Fourier Transform InfraRed spectroscopy with attenuated Total Reflectance Analysis (FTIR-ATR) (Bruker Hyperion 1000, Billerica, Massachusetts, USA) in the wave range of 3100–400 cm^−1^ was used to analyze the changes in the bonds of the PU foam (PU) at the end of the experiment (17 days) compared to the original polyurethane foam (PU-virgin). The foam was previously washed with distilled water (×3) and dried in an oven for 24 h at 80 °C.

#### 2.2.3. Analysis of the PU Foam with Thermogravimetric Analysis (TGA)

This analysis was performed using an SDT Q600 Thermogravimetric analyzer, Waters, TA instruments, Milford Massachusetts (USA) to characterize the changes in the thermal properties of the PU foam (PU) at the end of the experiment (17 days) compared to the original polyurethane foam (PU-virgin). The TGA was performed at 10 °C min^−1^ from 30 °C to 700 °C in a nitrogen atmosphere (flow rate 25 mL min^−1^) in 2–3 mg of polyurethane foam.

#### 2.2.4. Analysis of the PU Foam with Scanning Electron Microphotography (SEM)

The PU foam (PU) and original PU foam (PU-virgin) were analyzed after 10 and 17 days as feed for the mealworms by Scanning Electron Microscopy (SEM) using FEI TENEO New York (USA). The analysis was developed following [14]. Previously, the PU foam was washed by immersion in 0.88% (*w*/*v*) sodium hypochlorite for 2 h to eliminate any possible remains of microorganisms in the foam. Later, it was washed in triplicate in 100 mL of distilled water, stirred for 2 min at 150 rpm, and dried for 24 h at 80 °C and coated in platinum.

### 2.3. Gut Microbiome Analysis

#### 2.3.1. Enzyme Activities

The enzymatic activity of the esterases, lipases, proteases, and laccases able to break some of the specific bonds that form polyurethane molecules was measured in the gut of the mealworm larvae fed with PU foam and bran after 3, 10, and 17 days. The gut was obtained by dissecting three larvae previously washed with 2 mL of distilled water and air-dried. Once dissected, the gut was immersed in 1 mL of 0.1 M phosphate buffer (pH 7) and shaken. Subsequently, the homogenate was centrifuged (Eppendorf, MiniSpin, Hamburg, Germany) at 14,100× *g* for 10 min. The supernatant was diluted 1:10, and an enzyme activity analysis was carried out (sample). All the enzyme activity was analyzed on a GeneQuant 1300 spectrophotometer, VWR, Radnor, PA, USA. ***Protease activity*** was determined spectrophotometrically by hydrolysis of p-nitroaniline following the modified method of Preiser et al. [23]. A sample of 100 µL was mixed with 700 µL of reaction mixture and incubated for 90 min at 37 °C. The reaction was stopped with 800 µL of 30% (*v*/*v*) acetic acid, and the color change was measured at λ = 410 nm. The reaction mixture contained 0.05 M glycine Na–OH buffer (pH 10) and 0.001 M BAPNA dissolved in 1000 µL of DMSO. The control samples were analyzed in the same way, but the sample was replaced with a glycine Na–OH buffer. ***Esterase activity*** was determined spectrophotometrically by hydrolysis of p-nitrophenyl acetate (p-NPA) following the modified method of Oceguera-Cervantes et al. [24]. In a final volume of 1 mL, 100 µL of sample was mixed with 800 µL of sodium phosphate buffer (0.05 M; pH 6.5) and 100 µL of p-NPA solution in acetonitrile (0.01 M). The samples were incubated at 37 °C for 20 min. The reaction was stopped by placing them in an ice bath for 5 min. Subsequently, the samples were centrifuged at 10,000× *g* for 5 min and measured at λ = 410 nm. The control samples were analyzed in the same way, but the sample was replaced with a sodium phosphate buffer. ***Lipase activity*** was determined spectrophotometrically with the hydrolysis of p-nitrophenyl laurate (p-NPL) using a modification of the Kilcawley et al. [13] method. In a final volume of 2 mL, 100 µL of sample was mixed with 1.9 mL of sodium phosphate buffer (0.1 M; pH 7). The samples were incubated at 37 °C for 30 min. The reaction was stopped by placing them in an ice bath for 5 min and adding 0.85 mL of NaOH (0.5 M). Subsequently, the samples were centrifuged at 10,000× *g* for 5 min and measured at λ = 400 nm. The control samples were analyzed in the same way, but the sample was replaced with a sodium phosphate buffer. ***Laccase activity*** was measured using spectrophotometry following the modified method of Dhakar and Pandey [25]. The following were mixed and incubated for 1 min: 1 mL of final volume, 100 µL of sample, 800 µL of sodium acetate buffer (0.2 M/0.1 M; pH 4.5), and 100 µL of ABTS (0.01 M) dissolved in a sodium acetate buffer (0.2 M/0.1 M; pH 4.5). Subsequently, the samples were centrifuged at 10,000× *g* for 5 min and measured at λ = 420 nm. The control samples were analyzed in the same way, but the samples were replaced with a sodium acetate buffer.

ε The calculations of each activity were carried out using the molar extinction coefficients. Protease activity was calculated using the molar extinction coefficient at λ = 410 nm by Ƹ = 8.8 mM^−1^ cm^−1^ [26]; esterase activity at λ = 410 nm by ε = 18.5 mM^−1^ cm^−1^ [27]; lipase activity by λ = 400 nm ε = 14.8 mM^−1^ cm^−1^ [13], and laccase activity at λ = 420 nm by ε = 36 mM^−1^ cm^−1^ [25]. All the activities were expressed in µM^−1^ g^−1^ min^−1^.

#### 2.3.2. Gut DNA Extraction and Amplicon Sequencing

The DNA from the gut of mealworms fed with PU foam (PU) and wheat bran (bran) for 3, 10, and 17 days was extracted from the gut of four mealworms from the same feed container pooled to eliminate individual variability. The gut was harvested and added to a tube with 100 µL of phosphate buffer (0.1 M). The DNA was extracted using the Dneasy PowerSoil kit (Qiagen, Hilden, Germany). The quantity and quality of the DNA extracts were evaluated using a Qubit 3.0 Fluorometer (Invitrogen, Thermo Fisher Scientific, Waltham, MA, USA). The samples for sequencing were analyzed with the Illumina Nextera barcoded two-step PCR libraries (V4, ITS2) and sequenced on an Illumina MiSeq, v3, 2 × 300 bp. Demultiplexing and trimming the Illumina adaptor residuals and trimming the locus specific primer sequences were removed (Microsynth AG, Balgach, Switzerland).

#### 2.3.3. Bioinformatic Analysis

The analysis of the sequences was carried out as follows: the samples were received in fastq, entered into QIIME2 [28], and denoised using the dada2 algorithm [29], taking into account the replicates of each treatment to avoid their slight variations. Once the ASV (Amplicon Sequence Variant) was obtained, the taxonomic classification was performed using consensus-vsearch with the Silva 132 database (released in 2020) as a reference. The eukaryotes, archeas, mitochondria, chloroplasts, and non-assignments were eliminated before the statistical analysis. Finally, normalization was conducted through rarefaction, bringing all the samples to the same value using the value of the last sample that conserved a minimum of 3000 readings. The sequences were deposited in the DNA database with the accession code PRJEB54959.

### 2.4. Statistical Analysis

The results were analyzed using Statgraphics 18 and plotted through Sigmaplot v12.0. The correlations were performed using Spearman’s correlation test, and the statistical significance was determined using a two-way analysis of variance (ANOVA) with a post-hoc Least Significant Difference (LSD) Fisher Test. To study the microbial community, an NMDS was performed using the Bray–Curtis distance with the vegan package and the graphs were made using the ggplot2 package.

## 3. Results and Discussion

### 3.1. PU Foam Consumption and Its Effect on Mealworms

To the best of our knowledge, there are only a few reports about the efficiency of PU consumption by mealworms. The PU consumption by mealworm larvae was linear, showing a lower slope (y_1_ = 2.995 + 1.893x) than the bran consumption (y_2_ = 16.424 + 8.049x) (Appendix A). This reflected a lower PU consumption (35%) than the bran (100%), and a higher mealworm weight loss (86%) than the bran (97%) (Table 1).

This could be because the bran contains enough nutrients for the *T. molitor* mealworms’ metabolism [15], while PU, as the only source of carbon, does not provide sufficient nutrients to support growth. This was observed by Lou et al. [30] with polyethylene (PE) and polystyrene (PS). It could also be due to the high energy cost of eliminating the toxic compounds derived from the PU foam degradation [21]. Bulak et al. [22] observed 45% of PU consumption and 26–28% of mealworm weight loss after 58 days. If our experiment had lasted as long as Bulak et al.’s [22], we would have found similar data although mealworm death or metabolic exhaustion could occur [21]. Yang et al. [31] showed that mealworms fed with polystyrene were still capable of successful pupation, which indirectly proves that digestion and assimilation occurred after 32 days, although they had a lower fat content than those who ingested a conventional diet.

The results showed that PU is not as palatable for mealworms as bran, although they have been shown to decompose it [32]. However, according to Peng et al. [15], a higher degradation of polystyrene was observed with a mix of bran or corn flour and polystyrene.

### 3.2. Evidence of PU Foam Biodegradation by Mealworms

Evidence of changes in the functional groups of the PU used to feed the mealworms compared to the PU-virgin was provided by FTIR analysis at the end of the incubation period (17 days) (Figure 1). The main changes observed were in the peak intensity compared to the spike appearance and disappearance (functional groups). The PU showed less intensity in the spectrum peaks, such as 1090–1099 cm^−1^ (C–O–C bond), 1220–1225 cm^−1^ (C–N bond), 1536 (N–H bond), 1630–1736 (C=O bond), 2867–2916 (CH2), and 3288 cm^−1^ (–OH bond). The only signal where the PU-virgin was higher than the PU was at 2930–2940 cm^−1^, corresponding to the symmetrical and asymmetrical stretching vibrations of the CH bonds in the CH2 groups [33]. This general pattern demonstrates that the PU molecule was occluded, and only the external parts could be degraded by microorganisms and extracellular enzymes [32] without distinguishing the hard segments, such as the free aromatic bond breaking [12] or urethane ester/ether, from the soft ones, such as the plane urethane represented by C–N of N–H [32,34].

The TGA analysis also provided evidence of modifications in the structure of the PU used to feed the mealworms compared to the PU-virgin after 17 days (Figure 2). The PU weight loss was higher (97.5%) than the PU-virgin (94%). This could be attributed to the lower amount of soft and hard segments found in the PU with higher thermic degradation resistance [7]. The weight loss occurred in three phases: the first stage was from 220 to 280 °C, where about 30% of the weight loss occurred. This could be due to the release of volatile organic compounds [35,36]. The second stage was from 300 to 400 °C, with a weight loss of 60%, corresponding to hard and soft urethane segment dissociation [37]. The third stage was above 400 °C, where a lower fraction of PU (around 10%) was degraded, corresponding to the organic residue decomposition [36] (Figure 2).

The SEM also demonstrated the physical degradation of the PU after 10 and 17 days compared to the PU-virgin (Figure 3). The surface of the PU-virgin showed smooth edges with no apparent breaks (Figure 3A), while the PU used to feed the mealworms showed wrinkled edges and pits, cracking and erosion that could be attributed to mealworm chewing [32] (Figure 3B–D). Similar results were observed by Bulak et al. [22] and Khan et al. [38] on PU films exposed to *Aspergillus tubingensis*.

### 3.3. The Mealworm Gut with PU Consumption: The Enzyme Activity and Microbial Community Effect

#### 3.3.1. Enzyme Activity in the Mealworm Gut

The different enzymatic activity, such as lipases, esterases, proteases, and laccases associated with the polyurethane hydrolysis [39] was measured in the gut of mealworm larvae for both diets, the PU and the bran (Figure 4). These enzymes showed a significant (*p* ≤ 0.001) interaction between the type of diet and the sampling time. Enzyme activity in the bran diet was significantly higher than that in the PU diet throughout the experiment, with two apparent phases (Figure 4). Phase one was from 1 to 6 days, with almost constant values, and phase two was from 6 to 17 days, when the enzymatic activity tended to increase. This could be due to the bran depletion from the mealworm consumption. The mealworms thus had fewer available nutrients and a greater need for enzyme synthesis to obtain nutrients from the scarce food available. However, for the PU diet, the behavior was different. The values of the enzyme activity were lower and mostly constant throughout the experiment (Figure 4). This could be explained by the scarce availability of nutrients since the first sampling time, not permitting the synthesis of the required digestive enzymes to degrade the polyurethane [40]. The enzymes esterase and lipase showed higher values (Figure 4A,B) than protease and laccase (Figure 4C,D), probably because the former degraded any PU bond [11]. This low enzyme activity on PU could also be due to the synthesis of the corona protein, which inhibits the absorption of nanoparticles by intestinal cells, such as nanoplastics [41], or the synthesis of enzymes and molecules related to the immune system [42,43].

It has previously been demonstrated than microbiota and the secreted enzymes in the mealworm gut could be adapted to new diets, even poor ones with low nutrient availability, like the PU diet [44]. It has also been shown that one of the survival mechanisms of some insects in situations of stress or nutritional deficit is to consume their lipid reserves [45]. In addition, an increase in proteases inside insects only occurs as a last resort in extremely stressful situations since this would lead to protein biodegradation, which is the last element to degrade [46].

#### 3.3.2. The Mealworm Gut Microbial Community

The Illumina MiSeq analysis of PCR-amplified 16S rRNA fragments was used to assess the changes in the gut microbiome community of the mealworms fed PU throughout the experiment since the gut microbiome of insects has an important role to play in their digestion process [47]. The mealworm gut microbiome diversity for the PU diet was higher (average 3.23) than for the bran diet (average 2.90) (Figure 5). Similar results were observed by Wang et al. [32] and Peng et al. [15]. The Shannon diversity for the PU diet slightly increased throughout the experiment, probably due to changes in the proportion of microorganisms capable of degrading PU, while for the bran diet, it slightly decreased.

A non-metrical multidimensional scaling (NMDs) exhibited differences in the gut microbiome for the PU diet and the bran diet mainly after 10 and 17 days (Figure 6). The principal phylum observed for both diets (PU and bran) were Tenericutes, Protobacteria, and Firmicutes (Figure 7). Similar results have been found by other authors with different types of plastics [15,16,17]. Tenericutes was the dominant phylum, reaching the highest proportion for the bran diet (63%), while for the PU diet, it reached 51%. These proportions decreased throughout the experiment. At the end of the experiment (17 days), the opposite occurred, with Tenericutes higher for the PU diet (around 30%) than for the bran diet (17%). Nevertheless, the relative abundance of Firmicutes increased by 18% for the PU diet compared to 10% for the bran diet at the end of the experiment (17 days). Bacteroidetes were lower for the PU diet [48].

An analysis of the relative abundance at the genus level (Figure 7) indicated four dominant genera for both diets: *Spiroplasma* (average 66.5%), *Enterococcus* (11.08%), *Lactococcus* (9.75%), and *Pediococcus* (5.32%). *Spiroplasma* (Tenericutes) decreased throughout the experiment for the bran diet (7%) and for the PU diet (44%). At the end of the experiment, it was more abundant for the bran diet (69.49%) than for the PU diet (38.62%). *Spiroplasma* was also observed in many studies with different types of plastics [16,20] and it is considered a pathogen or a male-killing bacterium, but in the gut of mealworms, it is not harmful [49].

*Lactococcus* was the only bacteria that could be associated with the PU diet throughout the whole experiment (Figure 8). *Lactococcus* and *Pediococcus* (Firmicutes) (associated with the PU diet at 3–17 days) are lactic acid bacteria, which may have contributed to adjusting and maintaining the health of the gut microbiome [50]. *Lactococcus* and *Enterococcus* (Firmicutes) (associated with the PU diet at 3–17 days) (Figure 8) are also common insect gut bacteria and are known members of the *T. molitor* gut microbiome [20,51]. According to Lou et al. [30], understanding the approximate locations of different bacteria (*Lactococcus* was present in every part of the gut, while *Enterococcus* was absent in the foregut and anterior midgut [51]) is a good way to infer possible degradation pathways in the mealworm gut since *Enterococcus* is related to PU degradation [32].

At the end of the experiment (17 days), in addition to *Lactococcus*, *Pediococcus,* and *Enterococcus,* different bacteria could also be associated with the PU diet (Figure 8) such as *Paraclostridium* (Firmicutes), *Chryseobacterium* (Bacteroidetes), *Kosakonia*, and *Pseudomonas* (Proteobacteria). *Chryseobacterium* was observed in the mealworm gut with the different PS diets [52]. *Kosakonia,* a member of the Enterobacteriaceae family has been observed in PE and PS degradation [53,54,55]. *Pseudomonas* has also been associated with PS biodegradation [16,56]. The digestion process in the intestine of mealworms is more complex than it seems, and the role of whole microbiota and synergic interactions are important in PU degradation [16].

## 4. Conclusions

From our study, we can conclude that the larvae of *T. molitor* were able to consume around 35% of the PU. Larvae chewing increased the surface PU facilitating the mealworm gut microbiome and extracellular enzymes on the PU bond. *Lactococcus* was the only bacteria that could be associated with the PU diet throughout the whole experiment, although other microorganisms, such as *Paraclostridium* (Firmicutes), *Chryseobacterium* (Bacteroidetes), *Kosakonia*, and *Pseudomonas* (Proteobacteria) could also be associated with PU degradation. The above-mentioned PU biodegradation was demonstrated through structural and physical approaches such as FTIR, TGA, and SEM analysis.

## Figures and Tables

**Figure 1 polymers-15-00204-f001:**
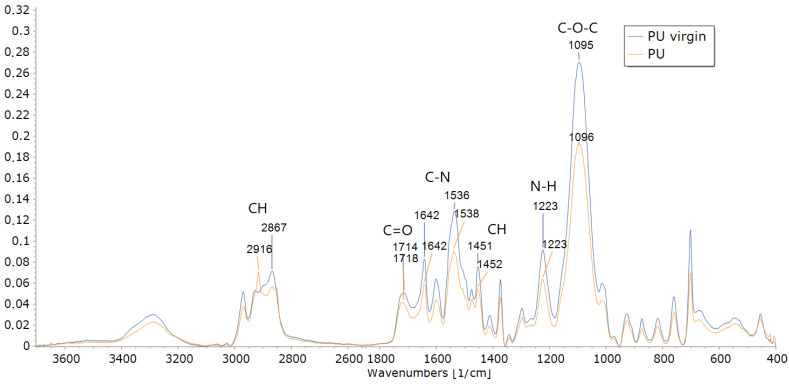
The Fourier Transform Infrared spectroscopy (FTIR) analysis of PU foam (PU) and original PU foam (PU-virgin) at the end of the experiment (17 days).

**Figure 2 polymers-15-00204-f002:**
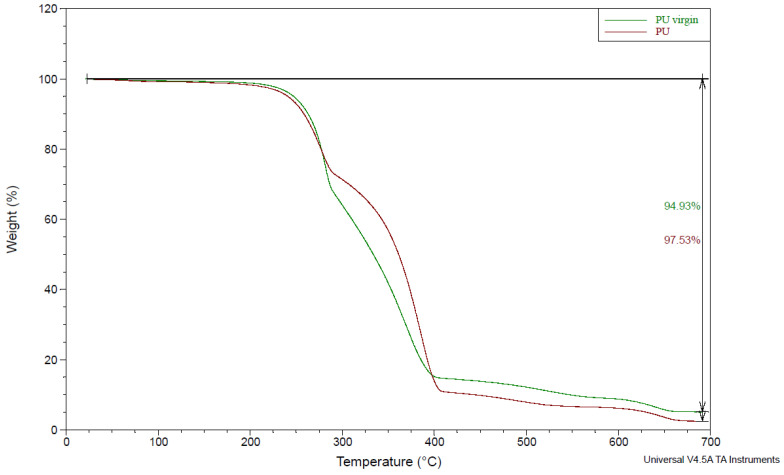
The Thermogravimetric analysis (TGA) of the PU foam (PU) and original PU foam (PU-virgin) at the end of the experiment (17 days).

**Figure 3 polymers-15-00204-f003:**
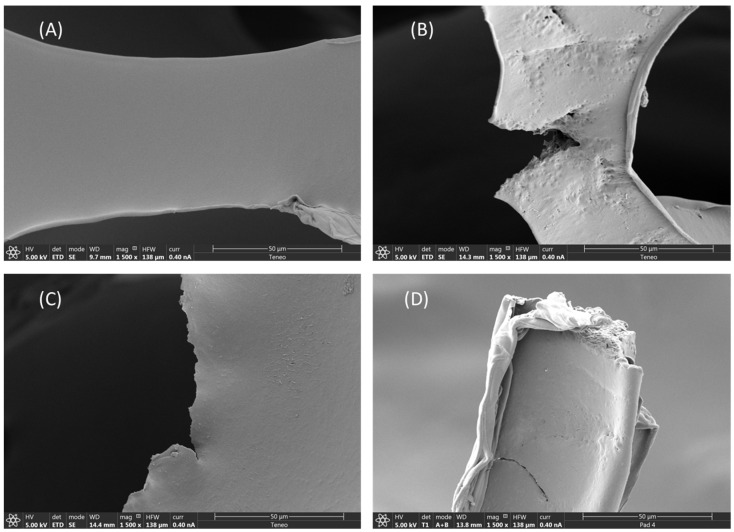
Scanning Electron Microphotography (SEM) of different PU foams: PU-virgin (**A**); PU after 10 days (**B**); PU after 17 days (**C**,**D**).

**Figure 4 polymers-15-00204-f004:**
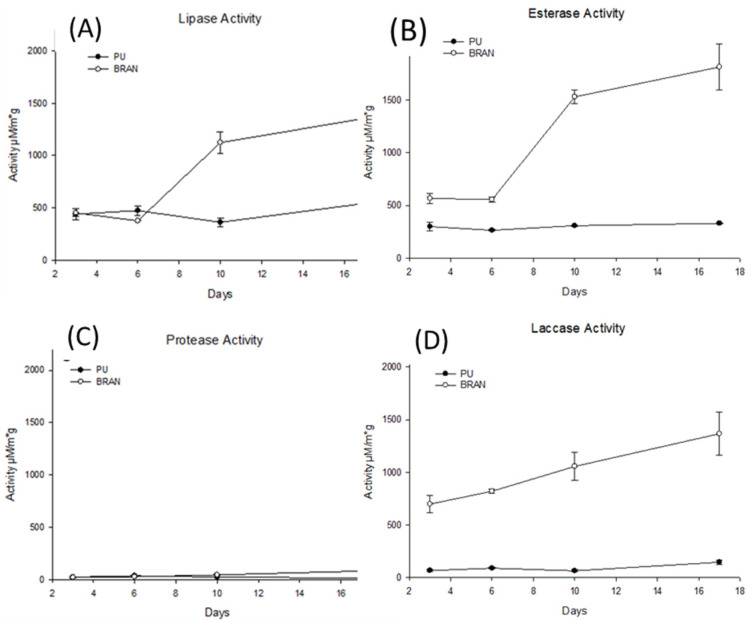
Polyurethane activity throughout the experiment for both PU and bran diets (**A**) lipase activity; (**B**) esterase activity; (**C**) protease activity; (**D**) laccase activity. For all the enzyme activity, a Two-Way ANOVA (Pdiet < 0.001); (Ptime < 0.001) was performed.

**Figure 5 polymers-15-00204-f005:**
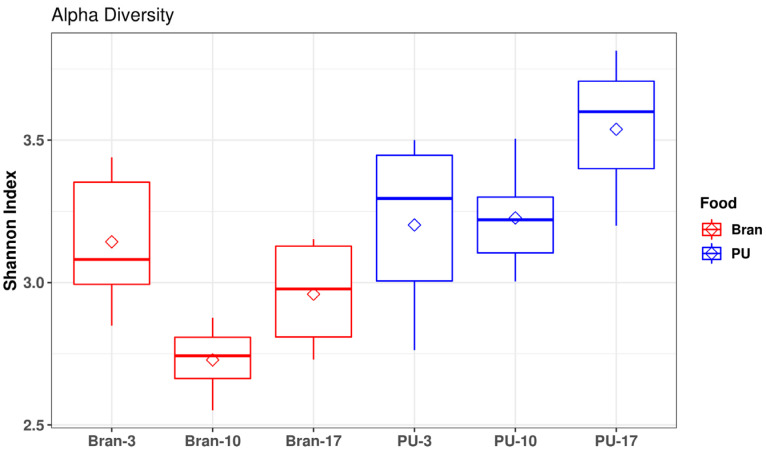
The Shannon diversity index of the gut microbiome of the PU and bran diets throughout the sampling (3, 10, and 17 days).

**Figure 6 polymers-15-00204-f006:**
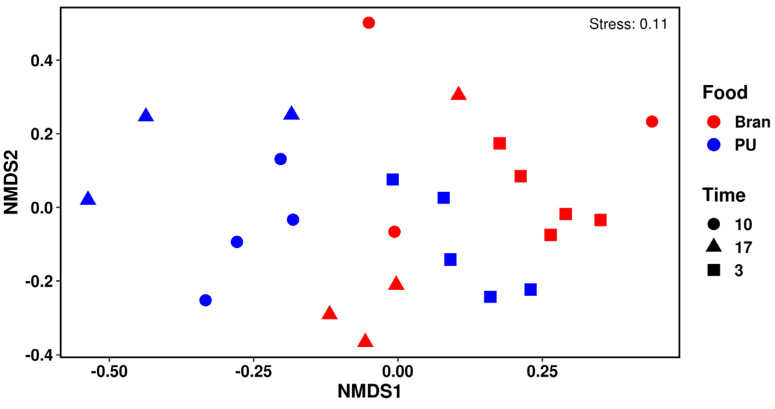
The NMDs of the gut microbial community for both diets (PU and bran) at different sampling times (3, 10, and 17 days).

**Figure 7 polymers-15-00204-f007:**
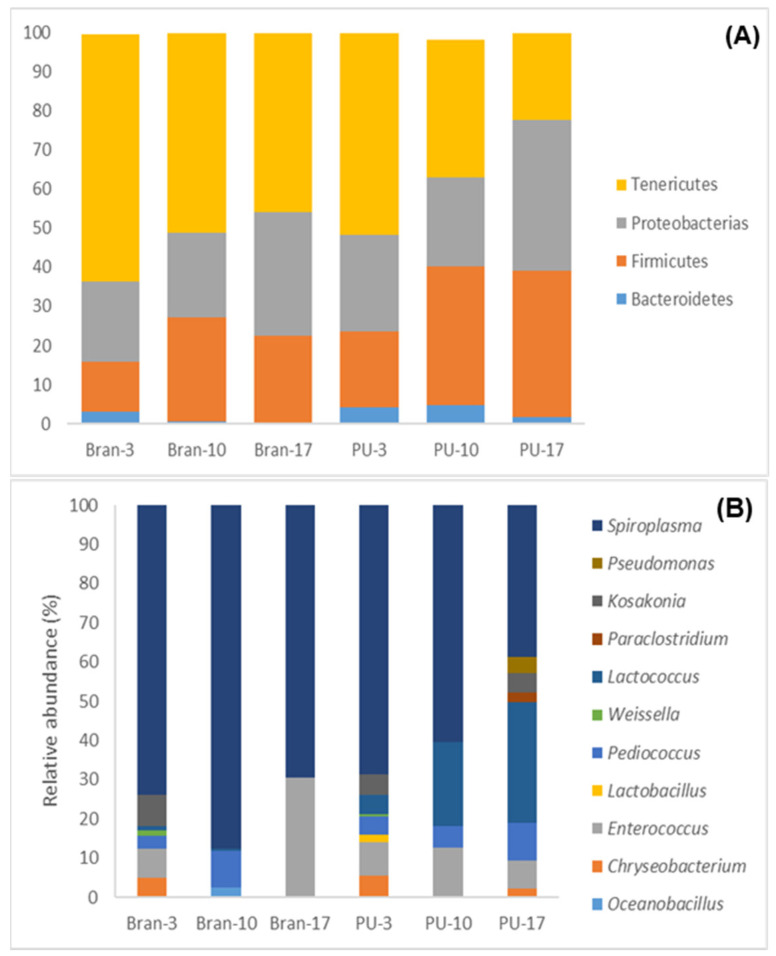
A bar plot of the community analysis of the mealworm gut for the PU diet (PU) and the bran diet (bran) with phylum (**A**) and genus (>1%) (**B**).

**Figure 8 polymers-15-00204-f008:**
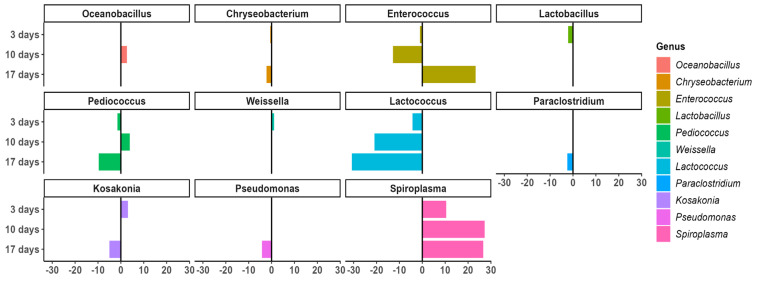
The levels and the differential abundance analysis of gut microorganisms for both diets.

**Table 1 polymers-15-00204-t001:** The mealworm larvae’s weight loss and feed consumption.

Days	Mealworm Weight Loss PU Diet (%)	Mealworm Weight Loss Bran Diet (%)	PU Consumption (%)	Bran Consumption (%)
0	100	100	0	0
3	97.95 ± 0.2	102.82 ± 1.35	8.27 ± 0.5	41.13 ± 1.04
6	95.32 ± 0.63	106.4 ± 1.12	14.35 ± 0.57	63.75 ± 5.35
10	93.53 ± 0.78	105.73 ± 4.36	22.73 ± 2.44	97.33 ± 0.79
17	85.84 ± 1.61	97.18 ± 7.82	34.78 ± 2.48	99.95 ± 0.05

## Data Availability

The data presented in this study are available on request from the corresponding author.

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
