# Peer review of "Polyurethane Foam Residue Biodegradation through the Tenebrio molitor Digestive Tract: Microbial Communities and Enzymatic Activity"

_polymers, 2022, doi:10.3390/polym15010204_

Round 1

Reviewer 1 Report

The manuscript by Orts et al. is nicely written and describes PU foam degradation through T. Molitor. The manuscript contains a number of characterization techniques to result in some conclusions. Therefore, I recommend acceptance of this manuscript after minor revision. 

Figure 4. The Lipase activity of PU foam goes down after 6 days and regenerate slightly only after approx. 14 days whereas, the Bran activity goes significantly higher than the PU foam after an initial period. The authors need to explain this phenomena with proper insights.

If Protease activity of PU goes down after 6 days, the authors need to explain the use of it. Application of PU under such circumstances is questionable.

Author Response

Figure 4. The Lipase activity of PU foam goes down after 6 days and regenerate slightly only after approx. 14 days whereas, the Bran activity goes significantly higher than the PU foam after an initial period. The authors need to explain this phenomena with proper insights.

The lipase activity in particular and the measured polyurethanases (Punase) in general, were significantly higher than those in the PU along the experiment (Line 266-270). Bran substrate showed higher PUnase than PU mostly constant along the experiment, that could be due to the low availability of nutrients (necessary for their life) and for the synthesis of digestive enzymes capable of degrading polyurethane (Line 277-280). No decrease at all during the assay was observed, although some variations were observed along time as the mentioned by Reviewer 1, but it cannot be thought that it was significant  due the polymeric and hydrophobicity of the polyurethane. Therefore, the difference should not attributed to any of the studied factors

If Protease activity of PU goes down after 6 days, the authors need to explain the use of it. Application of PU under such circumstances is questionable.

As  it was suggested by the other Reviewer (Reviewer 2), once all the enzymatic assays were at the same scale, it can be seen that protease was the lower activity in comparison to the others. From this new point of view, there is no significant variation in this activity, not providing the misunderstanding caused in the previous version.  We apologize for bringing Reviewer to wrong view

Reviewer 2 Report

This manuscript is mainly about the biodegradation of polyurethane foam residue through Tenebrio molitor digestive tract. With FTIR, TGA, SEM technologies, the authors evidenced that Tenebrio molitor larvae’s gut microbial communities and enzymes are involved in the degradation process. The PU consumption were estimated to be 35%. In addition, Lactococcus was identified as main bacteria for PU degradation while others like Firmicutes, Bacteroidetes, Proteobacteria were suggested as possible degrading bacteria. In general, the manuscript is of interest and may be accepted after completely addressing the following concerns.

(1) The scale bars in Figure 3A, 3B and 3C/3D are quite different with each other. It is hard for others to form a direct comparison.

(2) Are there any studies or guesses on which kind of enzymes in Lactoccus are involved in the biotransformation of PU?

(3) Can authors estimate and summarize a possible degradation mechanism of PU by Tenebrio molitor larvae?

Author Response

(1) The scale bars in Figure 3A, 3B and 3C/3D are quite different with each other. It is hard for others to form a direct comparison.

Done. They have been changed and Figure 3 A-D are in the same scale

(2) Are there any studies or guesses on which kind of enzymes in Lactoccus are involved in the biotransformation of PU?

There are some other studies that also showed that Lactococcus was involved in Tenebrio molitor larvae grown on xenobiotics such as plastics but there is a lack of INFO related to the main enzymatic routes in this specific microorganism . We are studying nowadays if this is or not a key microorganism for degrading this type of difficult substrates.

(3) Can authors estimate and summarize a possible degradation mechanism of PU by Tenebrio molitor larvae?

The estimation it is that there is a decrease on the C-O-C, C-N, C=O and CH2 that it can be attributed that PU was chewed by T molitor and this physical breakdown would permit an initial biological attack that it would be followed by breakdown chain that it is difficult to point out at the publication stage. We are studying the key microorganisms and metabolites, and once they will determine, a proposed chain mechanism would be proposed.